# Tensile Failure Behaviors and Theories of Carbon/Glass Hybrid Interlayer and Intralayer Composites

Weili Wu 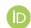

College of Textile Science and Engineering, Zhejiang Sci-Tech University, NO.928, 2nd Street, Qiantang District, Hangzhou 310018, China; wweili@zstu.edu.cn; Tel.: +86-150-2650-3803

**Abstract:** Hybrid composites combine various types of fiber that not only provide an effective method to minimize material costs but also enhance the mechanical properties of composites. The tensile fracture behaviors of hybrid composites are more complex than single-fiber composites due to various reinforcing fibers and hybrid effects, and the relationship between tensile behaviors and hybrid structures is not clear. In this paper, various structures of C/G (carbon/glass) interlayer and intralayer hybrid composites were designed, and tensile behaviors were investigated; it revealed that tensile failure is characterized by the synergistic effect and failure acceleration effect. Second, the tensile properties of interlayer and intralayer hybrid composites with various hybrid ratios and stacking structures were systematically analyzed; our results demonstrated that the tensile strength of interlayer and intralayer hybrid composites was predominantly impacted by the hybrid ratio of C/G and increased with the increase in carbon fiber content. For interlayer hybrid composites, with the assistance of the synergistic effect, excellent tensile strength could be obtained for the glass fiber sandwiched carbon fiber structure. For intralayer hybrid composites, the tensile strength was small, while the dispersion degree was high. We compared the tensile properties with theoretically calculated values based on the rule of mixing (ROM) and revealed that the tensile modulus and strength of interlayer and intralayer hybrid composites exhibited a positive hybrid effect. This work serves as a foundation for the structural optimization and potential applications of C/G non-crimp hybrid composites.

**Keywords:** carbon/glass hybrid composites; C/G hybrid ratio; tensile failure behaviors; synergistic effect; failure acceleration effect; hybrid effect





## 1. Introduction

Currently, high cost has been the most critical factor hindering the wider application of carbon fiber composites. Hybrid composites combine low-cost fiber (such as glass fiber) with high-cost fiber (such as carbon fiber, aramid fiber, and basalt fiber), which can not only reduce material cost but also allow the utilization of the complementary advantages of different fibers, enhancing the mechanical properties of composites [1–3]. For example, mixing carbon and glass fiber can make up for the brittleness and poor impact resistance of carbon fiber, and it can also improve the modulus and stiffness of glass fiber composites. Furthermore, combining carbon fiber and graphene yields a structure with an exceptional thermo–electro–mechanical property [4–6]. Therefore, research on hybrid composites has certain practical significance [7–9].

The tensile performance of composites is one of the most significant mechanical properties and can be improved by utilizing high-performance fibers [10]. It is determined by many factors, including the mechanical properties of its reinforcing fibers–matrix interface, fiber content, and extension state, etc. [11–13]. Recent studies focused on the influence of the fiber type and hybrid ratio on enhancing the tensile properties of hybrid composites [14–16]. Li et al. [17] investigated the effect of the hybrid ratio on the tensile properties of carbon/basalt fiber interlayer hybrid composites and revealed the mechanical properties

of hybrid composites improve with the increase in carbon fiber content and complied with ROM (rule of mix). Zhou et al. [18] reported the effect of the hybrid ratio on the tensile properties of carbon/glass intralayer hybrid composites and drew the same conclusion; however, this research was limited to only two hybrid structures. Chiang et al. [19] studied the effect of carbon/glass hybrid structures produced by the winding method on tensile properties and proved that hybrid composites complied well with a ROM, showing no obvious hybrid effect. However, Zeng et al. [20] changed the hybrid material into an aramid/carbon filament produced by the winding process and revealed that tensile properties decrease with the increase in carbon fiber content, presenting a negative hybrid effect.

The change in hybrid structure is another key factor that determines tensile properties and failure behaviors, and the hybrid structure is achieved by the alteration of the hybrid form and stacking sequence. The two main hybrid forms include the interlayer and intralayer hybrid, which are depicted in Figure 1. Various interlayer hybrid structures are produced by adjusting the stacking sequences of different types of reinforcement fabrics, while intralayer hybrid structures are obtained by altering the hybrid fabrics themselves. Recent studies have shown that hybrid structures have an influence on the mechanical properties of interlayer hybrid composites [21–23].

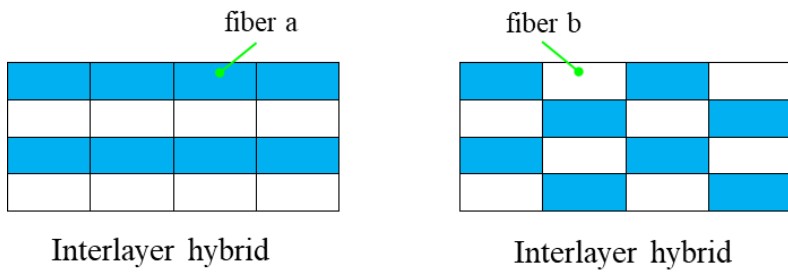

**Figure 1.** Hybrid form.

Li et al. [24] studied the effect of stacking sequences on the glass/ramie fiber mat-reinforced interlayer hybrid composites and observed that the distribution of glass and ramie fibers in the core and surface layers, respectively, resulted in a hybrid composite with 71% higher tensile strength than that of glass fiber composites. When the outer layer of ramie fiber breaks, the glass fiber in the core layer can effectively prevent the fracture of the ramie fiber and expansion of cracks attributed to hybrid composites exhibiting positive hybrid effects. Pandya et al. [25] investigated the sequence of carbon/glass woven interlayer hybrid composites, and their results showed that as glass and carbon fibers distribute in the surface and core layers, respectively, hybrid composites exhibit excellent tensile strength and failure strain; this was supported by Dong's work [26]. Ebrahimnezhad et al. [27] tested the tensile properties of carbon/kevlar/glass interlayer hybrid laminates and drew the same conclusion. Meanwhile, the microstructure investigation revealed the delamination and fibers imprint were dominant failure mechanisms for the hybrid composites. Khosravani et al. [28,29] investigated the mechanical responses and failure behaviors of honeycomb sandwich composites; the failure modes were featured by brittle fracture of the adhesive.

Regarding the intralayer structures, Alsaadi et al. [30] studied the intralayer carbon/aramid/glass woven fabric-reinforced composites and revealed the stacking sequence of the outer skin of laminates has a significant effect on the tensile modulus. In particular, glass fiber on the outside resulted in an increase in tensile strength by 60% relative to the full carbon fiber laminates. Karahan et al. [31] tested the tensile properties of composites made from carbon/aramid hybrid woven fabrics, and the effect of weaving structure and hybridization on the mechanical properties was investigated. It was found that the Young's modulus of the hybrid composites was 16%–63% higher than expected. Junior et al. [32] studied the tensile strength of intralayer hybrid ramie/cotton composites with various stacking sequence configurations, and their results showed that the main parameter governing the tensile properties of composites is the ramie volume fraction, while the contribution

of cotton fiber is minimal, the composites containing 45% ramie fibers displayed the highest tensile strength.

Mechanical properties of hybrid composites which consist of two or more types of fibers are more complex than those of single component material. The previous investigations on the tensile properties of hybrid composites primarily focused on interlayer hybrid structures or some intralayer structures with a simple form. However, there were few comprehensive reports on the tensile properties of intralayer hybrid composites with non-crimp fabrics (NCFs) that could achieve better mechanical properties. In this paper, we performed systematic research on the tensile properties and failure modes of carbon/glass interlayer and intralayer hybrid composites. First, the carbon/glass (C/G) interlayer and intralayer hybrid composites with various hybrid structures and mixed ratios were designed and the tensile properties were explored. Subsequently, two failure theories synergistic effect and failure acceleration effect were proposed to analyze the failure behaviors and tensile properties of interlayer and intralayer hybrid composites. This work laid the foundation for the research and application of C/G reinforced hybrid composites.

## 2. Materials and Methods

### 2.1. Materials

The experiment involved carbon/glass interlayer and intralayer hybrid composites reinforced with warp-knitted NCFs which perform excellent mechanical properties due to their straight fiber alignment. Carbon fiber was supplied by TORAY Inc. (Tokyo, Japan), glass fiber was purchased from CPIC glass fiber Inc. (Chongqing, China), and the epoxy resin was provided by SWANCOR Inc. (Shanghai, China). The parameters of fibers and resin are presented in Table 1.

**Table 1.** Parameters of raw materials.

| Materials | Sources | Density (Kg/m$^3$) | Tensile Strength (MPa) | Tensile Modulus (GPa) |
|---|---|---|---|---|
| Glass fiber | CPIC ECT469L-2400 | 2560 | 2366 | 78.7 |
| Carbon Fiber | TORAY 620SC-24K-50C | 1770 | 4175 | 234 |
| Epoxy Resin | SWANCOR 2511-1A/BS | 1100 | 73.5 | 3.1 |

Table 2 reports the specifications of NCFs utilized in the experiment, comprising a pure carbon fiber fabric, a glass fiber fabric, and four types of C/G hybrid fabrics. The structural schematic diagram of the fabrics is displayed in Figure 2.

**Table 2.** Specifications for hybrid fabrics.

| Fabric Type | Areal Density (g/m$^2$) | | Ratio of C/G |
|---|---|---|---|
| | Carbon Fiber | Glass Fiber | |
| Carbon fabric | 728.3 | 0 | 1:0 |
| Glass fabric | 0 | 944.9 | 0:1 |
| C-G | 364.2 | 472.4 | 1:1 |
| C-C-G-G | 364.2 | 472.4 | 1:1 |
| C-G-G | 242.8 | 629.9 | 1:2 |
| C-G-G-G-G | 145.7 | 755.9 | 1:4 |

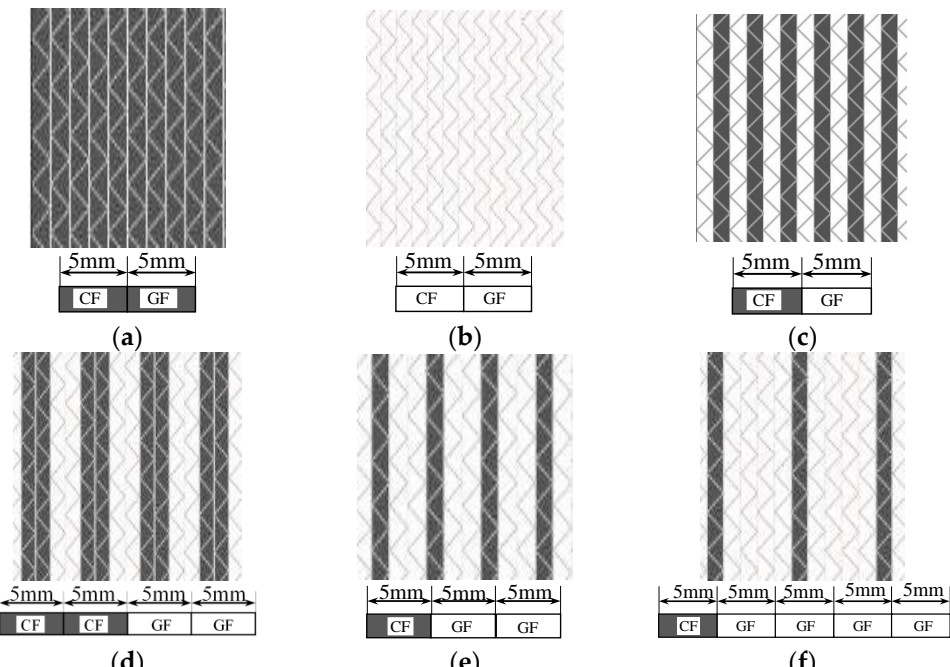

**Figure 2.** Schematic structures of six types of NCFs. (**a**) Carbon fabric; (**b**) glass fabric; (**c**) [C-G]; (**d**) [C-C-G-G]; (**e**) [C-G-G]; (**f**) [C-G-G-G-G].

The interlayer C/G hybrid composites were created utilizing different carbon fiber fabrics and glass fiber fabrics, varying the fabric sequences and C/G ratios to achieve different interlayer hybrid structures. Four types of carbon/glass hybrid fabrics, namely [C-G], [C-C-G-G], [C-G-G], and [C-G-G-G-G], were adopted to produce the intralayer composites structures.

### 2.1.1. Interlayer Hybrid Structures

Interlayer C/G hybrid composites are composed of glass and carbon fiber fabrics, and four C/G hybrid ratios were designed. Under the same hybrid ratio, different hybrid structures were produced by adjusting the stacking sequences of carbon NCFs and glass NCFs. Table 3 illustrates the interlayer hybrid scheme.

**Table 3.** Stacking configurations of interlayer hybrid structures.

| C/G Hybrid Ratio | Stacking Sequences | | | |
|---|---|---|---|---|
| 1:1 | [G/G/C/C] | [G/C/C/G] | [C/G/G/C] | [G/C/G/C] |
| 1:2 | [G/G/C] | [G/C/G] | | |
| 1:3 | [G/G/G/C] | [G/G/C/G] | | |
| 1:4 | [G/G/G/G/C] | [G/G/G/C/G] | [G/G/C/G/G] | |

Note: the carbon fabric layer is indicated by black color, while the white color denotes the presence of glass fabric in the laminates.

### 2.1.2. Intralayer Hybrid Structures

Table 4 presents the intralayer hybrid scheme consisting of four types of carbon/glass hybrid fabrics for producing the intralayer composite structures. These fabrics were arranged differently with the same hybrid ratio to achieve various dispersion degrees and structures.

**Table 4.** Stacking configurations of intralayer hybrid structures.

Note: The number in Table 4 corresponds to the dispersion degree of the intralayer composites, representing the extent of dislocation between the upper and lower layers. A higher dispersion degree means a more obvious fabric dislocation.

### 2.2. Tensile Testing

After manufacturing hybrid composites using the VARTM process, the laminates were subjected to tensile testing following the ASTM D3039 standard. The average value of five samples was adopted for data analysis, and the fiber volume fraction was maintained at 50%. The detailed parameters of specimens are presented in Table 5.

**Table 5.** Parameters of testing samples of hybrid composites.

| Hybrid Types | C/G Hybrid Ratios | Layup Number | Total Thickness /mm | Sample Width /mm |
|---|---|---|---|---|
| Carbon fiber fabric | 1:0 | 4 | 3.2 | 15 |
| Glass fiber fabric | 0:1 | 4 | 3.2 | 15 |
| Interlayer composites | 1:1 | 4 | 3.2 | 15 |
| | 1:2 | 3 | 2.4 | 15 |
| | 1:3 | 4 | 3.2 | 15 |
| | 1:4 | 5 | 4 | 15 |
| Intralayer composites | 1:1 | 4 | 3.2 | 10/20 |
| | 1:2 | 4 | 3.2 | 15 |
| | 1:4 | 4 | 3.2 | 25 |

The tensile experiment utilized the LD26.305 Universal material testing machine provided by Lambert Sansi Material Testing Co., Ltd. (Shenzhen, China), the testing

sample is displayed in Figure 3. According to the standard, the testing speed was set to 2 mm/min, and the tensile stress attenuation rate was set to 80% as the testing end parameter. During the testing, a camera was used to capture the specimen's failure process. The testing results of interlayer and intralayer hybrid composites were displayed in Tables 6 and 7, respectively.

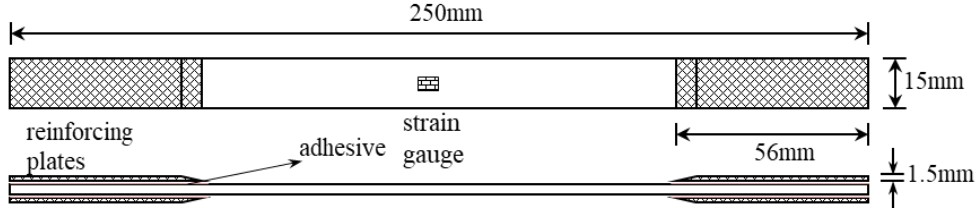

**Figure 3.** The testing sample.

**Table 6.** The testing results of interlayer hybrid composites.

| C/G Hybrid Ratios | Interlayer Structures | Tensile Modulus /Gpa | Error/Gpa | Tensile Strength /Mpa | Error/Mpa | Tensile Fracture Strain | Error |
|---|---|---|---|---|---|---|---|
| C:G = 0:1 | [G/G/G/G] | 38.56 | 0.30 | 863.40 | 22.08 | 2.25 | 0.06 |
| C:G = 1:4 | [G/G/G/G/C] | 51.71 | 0.69 | 873.68 | 30.89 | 1.62 | 0.06 |
| | [G/G/C/G/G] | 54.13 | 0.94 | 908.52 | 10.58 | 1.62 | 0.04 |
| | [G/G/G/C/G] | 53.06 | 1.24 | 925.91 | 42.14 | 1.76 | 0.08 |
| C:G = 1:3 | [G/G/G/C] | 55.18 | 1.16 | 932.23 | 18.82 | 1.70 | 0.03 |
| | [G/G/C/G] | 56.30 | 0.71 | 981.42 | 31.30 | 1.75 | 0.06 |
| C:G = 1:2 | [G/G/C] | 62.94 | 0.58 | 1024.49 | 36.78 | 1.64 | 0.06 |
| | [G/C/G] | 63.53 | 0.88 | 1098.44 | 37.55 | 1.69 | 0.09 |
| C:G = 1:1 | [G/G/C/C] | 76.02 | 0.99 | 1216.34 | 52.28 | 1.58 | 0.09 |
| | [C/G/G/C] | 75.11 | 0.30 | 1237.89 | 57.23 | 1.65 | 0.08 |
| | [G/C/G/C] | 76.46 | 1.92 | 1299.59 | 30.86 | 1.70 | 0.05 |
| | [C/G/G/C] | 75.67 | 1.19 | 1299.97 | 31.17 | 1.70 | 0.07 |
| C:G = 1:0 | [C/C/C/C] | 110.91 | 2.85 | 1606.44 | 28.86 | 1.45 | 0.03 |

**Table 7.** The testing results of intralayer hybrid composites.

| C/G Hybrid Ratios | Intralayer Structures | Tensile Modulus /Gpa | Error/Gpa | Tensile Strength /Mpa | Error/Mpa | Tensile Fracture Strain | Error |
|---|---|---|---|---|---|---|---|
| C:G = 1:4 | [C-G-G-G-G-0] | 51.30 | 2.55 | 942.69 | 24.86 | 1.82 | 0.04 |
| | [C-G-G-G-G-0.5] | 56.06 | 1.98 | 945.50 | 27.45 | 1.69 | 0.05 |
| | [C-G-G-G-G-1] | 53.27 | 0.20 | 922.35 | 20.42 | 1.73 | 0.04 |
| | [C-G-G-G-G-2] | 59.01 | 1.72 | 917.83 | 26.26 | 1.57 | 0.03 |
| | [C-G-G-G-G-1.5] | 53.54 | 1.05 | 911.04 | 37.56 | 1.71 | 0.07 |
| | [C-G-G-G-G-2.5] | 52.71 | 1.61 | 826.74 | 8.58 | 1.56 | 0.03 |
| C:G = 1:2 | [C-G-G-0] | 64.29 | 1.97 | 1087.71 | 23.73 | 1.69 | 0.04 |
| | [C-G-G-0.5] | 62.01 | 1.70 | 1029.45 | 40.13 | 1.66 | 0.08 |
| | [C-G-G-1] | 63.99 | 1.90 | 1073.21 | 29.09 | 1.68 | 0.05 |
| | [C-G-G-1.5] | 62.70 | 3.62 | 883.99 | 22.50 | 1.43 | 0.03 |
| C:G = 1:1 | [C-C-G-G-0] | 77.22 | 3.98 | 1262.06 | 39.82 | 1.68 | 0.02 |
| | [C-C-G-G-0.5] | 77.89 | 2.55 | 1315.33 | 39.49 | 1.69 | 0.04 |
| | [C-C-G-G-1] | 76.31 | 1.91 | 1274.12 | 53.10 | 1.71 | 0.07 |
| | [C-C-G-G-1.5] | 74.54 | 3.01 | 1286.36 | 43.57 | 1.73 | 0.05 |
| | [C-C-G-G-2] | 80.01 | 2.14 | 1197.94 | 48.33 | 1.50 | 0.06 |

### 3. Tensile Process and Failure Behaviors of Interlayer and Intralayer Hybrid Composites

Due to the significant release of fracture energy during the tensile process, it was difficult to collect the failure data accurately. Moreover, the strain gauges fall off from the testing samples, making it challenging to analyze the tensile failure behavior. Thus, the tensile force–displacement attenuation curves and failure process were used instead to analyze the tensile failure behaviors. Figure 4a exhibits the tensile force–displacement curves of pure-carbon fiber and glass fiber-reinforced composites. The curves indicate that the tensile force of both carbon fiber and glass fiber composites exhibited a cliff-like drop with only one failure. We attributed the large modulus of the carbon fiber, which stored high amounts of fracture energy previous to the failure (making the process more violent), to the sample's collapse failure; given this, we were unlikely to obtain a complete failure sample of carbon fiber composites. In contrast, the failure of the glass fiber sample was relatively complete, as shown in Figure 4b. The tensile failure of glass fiber composites was characterized by the fiber's uniform pull-out fracture.

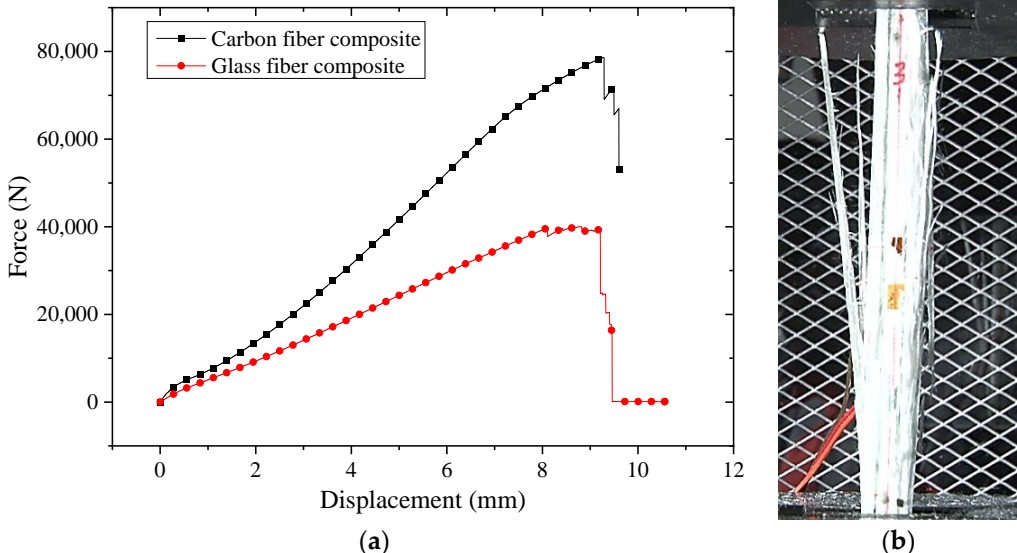

**Figure 4.** Tensile force–displacement curves of carbon fiber and glass fiber composites: (**a**) Tensile curves; (**b**) failure specimen of glass composites.

#### 3.1. Failure Behaviors of Interlayer and Intralayer Hybrid Composites with the Hybrid Ratio C:G = 1:1

Figure 5 shows the tensile force–displacement curves of interlayer hybrid composites with a hybrid ratio C:G = 1:1. As the specimen suffered from the tensile force due to the various failure modes of the two reinforcement materials, we observed that hybrid composites tended to exhibit various failure features. From Figure 5, for samples with equivalent carbon fiber and glass fiber content, the tensile force in the interlayer hybrid composites with a four-layer laminate increased linearly with the increase in tensile displacement. Once it reached the maximum load, the tensile force decreased due to first-order failure, followed by a slight rise and decline until complete failure (also known as second-order failure). During tension, the carbon fiber failed due to its low strain attributed to the first-order failure, and the glass fiber continued to assume the residual load and was quickly damaged due to its insufficient modulus, leading to second-order damage.

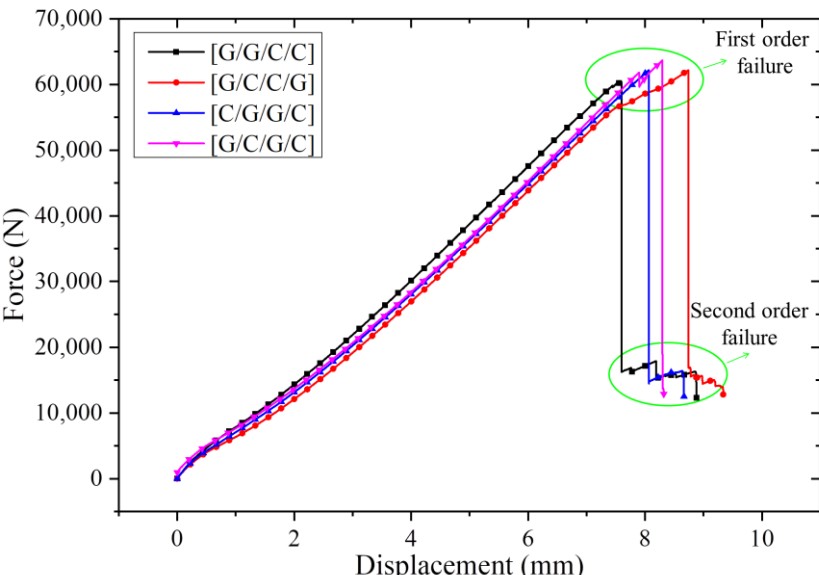

**Figure 5.** Tensile force–displacement curves of interlayer hybrid composites with C:G = 1:1.

Figure 6a shows the failure specimen of an asymmetric interlayer structure [G/G/C/C]. During tension, first, the carbon fiber failed and exhibited first-order failure. Because the carbon fiber content was 50%, it carried most of the tension force due to its high modulus in the initial phase, resulting in the high damage energy attenuation after failure that caused the glass fiber to suffer from collapse damage. Afterward, the whole specimen quickly failed while the glass fiber loaded the residual force, as shown in the second-order failure. Figure 6b presents the interlayer hybrid structure with glass fiber sandwiching carbon fiber [G/C/C/G] in which the second-order failure did not appear, but the first-order failure was obvious, which was caused by the damage acceleration effect. First, the carbon fiber in the core layer was the first to fail, leading to rapid failure of the entire sample due to the energy released by the carbon fiber which resulted in damage to the glass fiber.

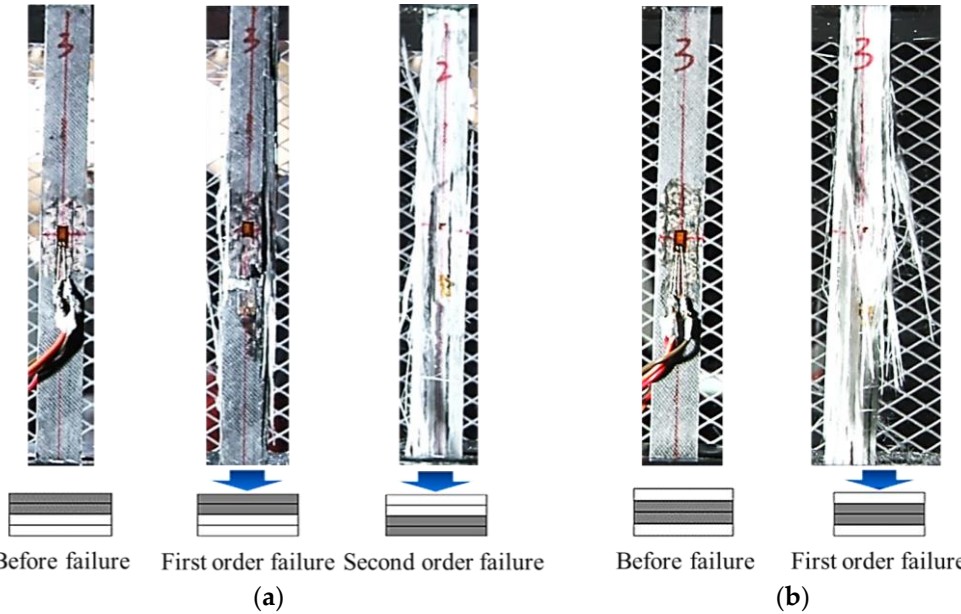

**Figure 6.** Failure process of the interlayer hybrid composites: (**a**) [G/G/C/C]; (**b**) [G/C/C/G]. Note: The arrow refers to the observation surface of the samples.

Figure 7a shows the tensile force–displacement curves of intralayer hybrid composites with a hybrid ratio of C:G = 1:1. The results indicated that the tensile curves of the intralayer and interlayer hybrid composites were similar. Figure 7b shows that carbon fiber broke first under tension, resulting in first-order failure. However, due to the energy absorption and force sharing by the glass fiber, the carbon fiber did not suffer from the crushed failure, leading to weaker damage in the glass fiber. As the force continued to increase, glass fiber tended to fail quickly and exhibit second-order failure. From the tensile curves, while the dispersion degree of C/G intralayer hybrid composites such as [C-C-G-G]-1.5 and [C-C-G-G]-2 was relatively high, the failure process of samples exhibited violence, with only first-order failure occurring.

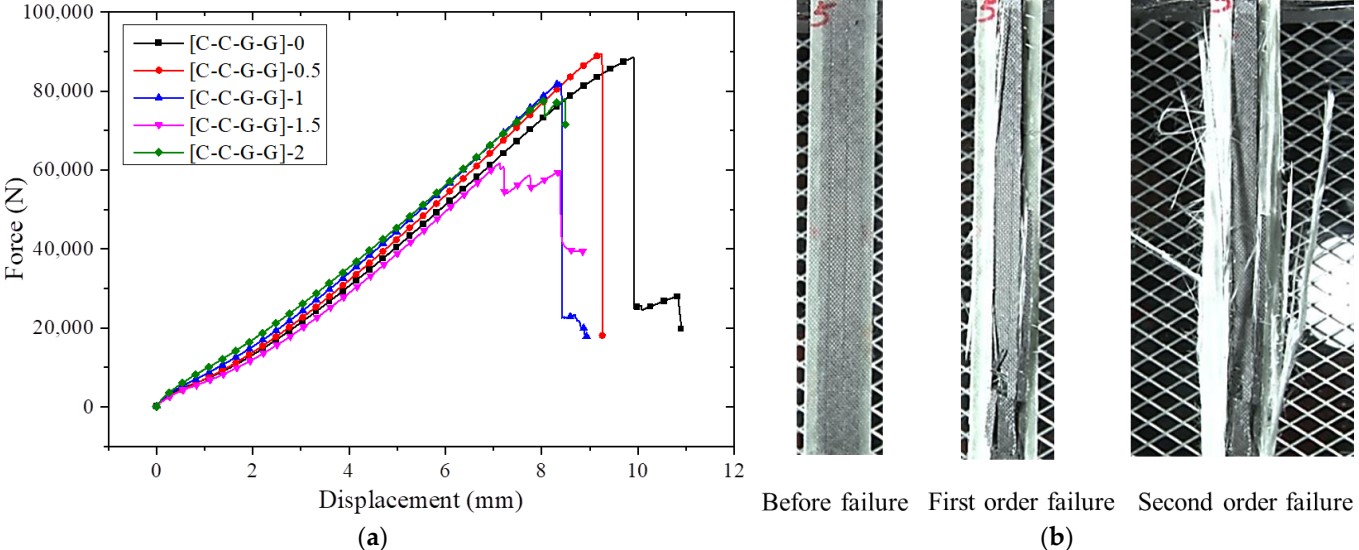

**Figure 7.** Tensile properties of intralayer hybrid composites with C:G = 1:1. (**a**) Tensile force–displacement curves; (**b**) tensile failure process.

### 3.2. Failure Behaviors of Interlayer and Intralayer Hybrid Composites with the Hybrid Ratio C:G = 1:2

Figure 8 shows the tensile force–displacement curves of interlayer and intralayer hybrid composites with hybrid ratios of C:G = 1:2. Compared with Figures 5 and 7a, with the increase in glass fiber content, the failure behaviors of hybrid composites were almost the same, and the first-order force drop occurred after the sample reached the maximum load. However, after the first-order force decay, small amplitude and multi-step force fluctuations appeared, and the second-order failure process remained in a longer phase. This was primarily because the carbon fiber content fell, reducing the damage sustained by glass fiber in the first-order failure; as a result, glass fiber composites could assume a higher force in the second-order failure. Furthermore, the failure process of the interlayer structure [G/C/G] displayed a short second-order failure process, indicating crush damage, while the second-order failure process of [G/G/C] lasted longer. For the structure with glass fiber sandwiching carbon fiber, carbon fiber was destroyed first, causing the first-order failure and transferring most of the damage energy to the outer glass fiber layer, causing severe damage to the glass fiber and accelerating its failure. Thus, the second-order failure related to the damage acceleration effect occurs more quickly. Figure 9a shows the fracture of the interlayer structure [G/C/G], with glass fiber sandwiching carbon fiber, and after the first-order failure, the glass fiber exhibited a certain amount of damage.

By comparing C:G = 1:2 with C:G = 1:1, the failure process of intralayer hybrid composites was proved to be similar, as the dispersion degree increased, the failure process became more violent; therefore, [C-G-G-1.5] and [C-G-G-1] hardly presented secondary destruction. The failure specimen of the intralayer structure [C/G/G-1] is shown in Figure 9b. After

undergoing first-order failure, the integrity of the glass fiber specimen remained high; later, the specimen underwent a second-order failure caused by the glass fiber.

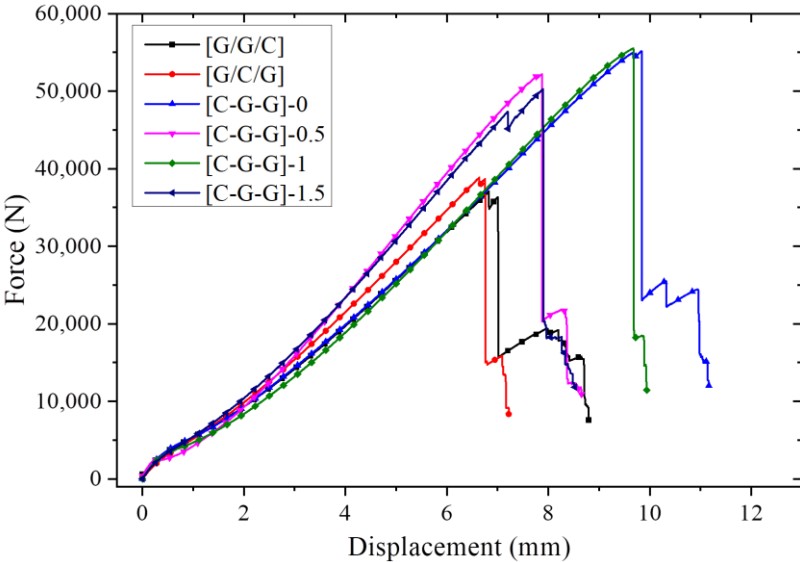

**Figure 8.** Tensile force–displacement curves of intralayer hybrid composites with the hybrid ratio C:G = 1:2.

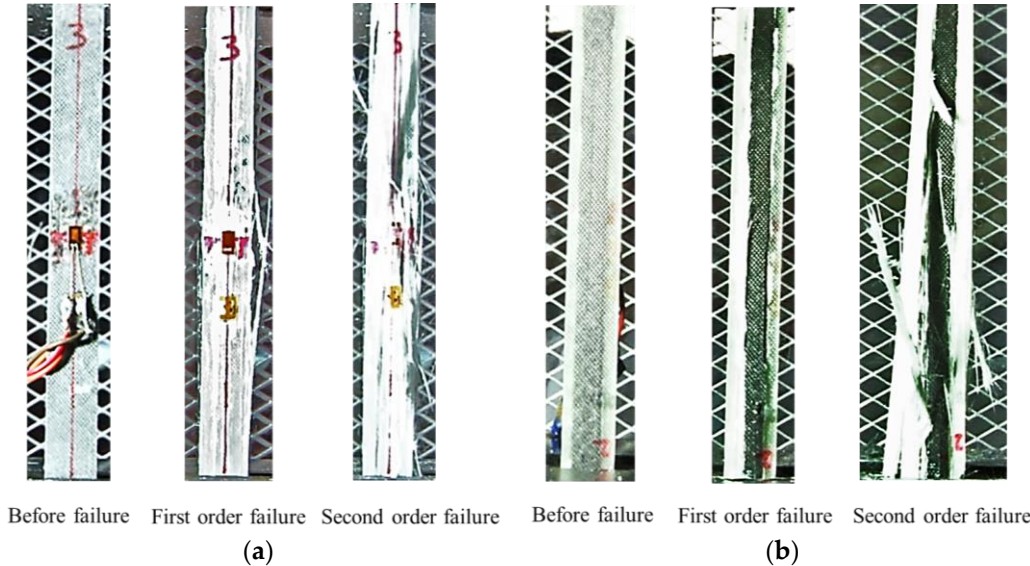

**Figure 9.** Failure process of hybrid composites: (**a**) Interlayer hybrid structure [G/C/G]; (**b**) intralayer hybrid structure.

### 3.3. Failure Behaviors of Interlayer and Intralayer Hybrid Composites with the Hybrid Ratio C:G = 1:3

Figure 10 shows the tensile force–displacement curves of interlayer hybrid composites with a hybrid ratio of C:G = 1:3. The tensile force of C:G = 1:3 fell sharply in the first phase, and then it presented a slight rise followed by a drop until the samples completely failed. Compared to the hybrid ratios C:G = 1:1 and 1:2, the second-order failure force was closer to the first maximum load. Figure 11a shows the specimen of the interlayer structure [G/G/C/G]. The failure of carbon fiber caused glass fiber damage to a certain extent, and then the second-order failure occurred as the tensile elongation reached the glass fiber's strain limit. Figure 11b presents the failure process of the asymmetric structure [G/G/G/C].

After the first failure of the carbon fiber, the damage to the glass fiber was low, and the specimen damage was comparatively complete due to a weak damage acceleration effect.

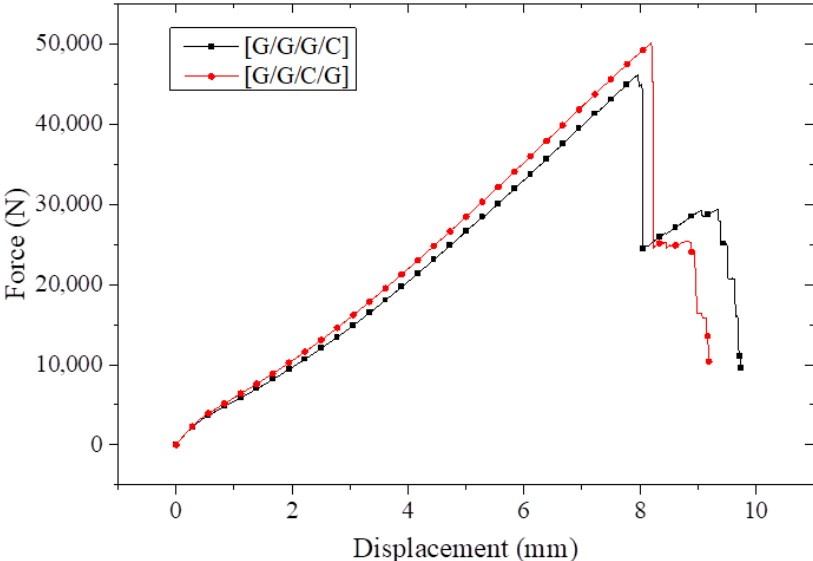

**Figure 10.** Tensile force–displacement curves of interlayer composites with the hybrid ratio C:G = 1:3.

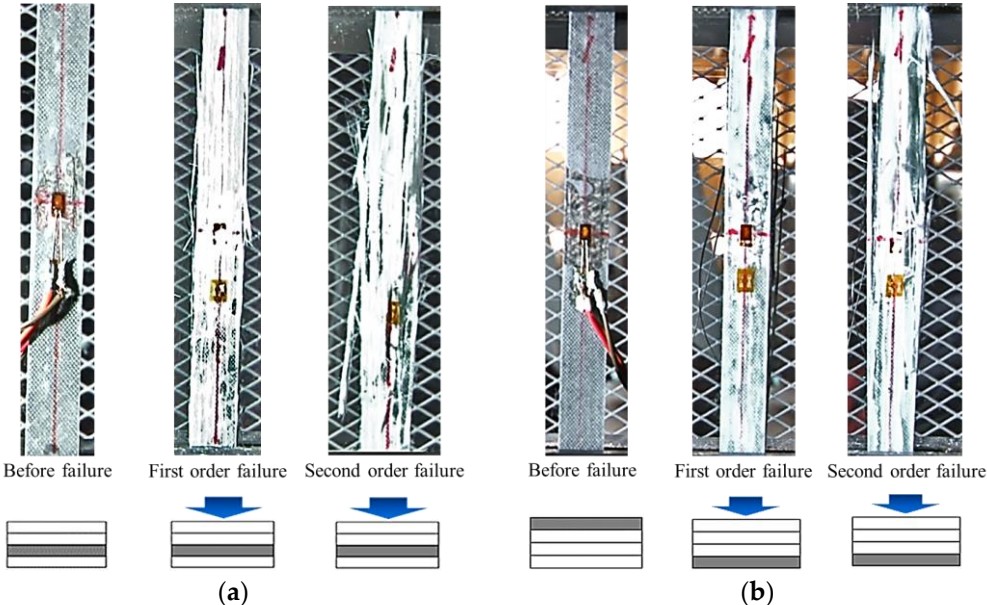

**Figure 11.** Failure process of interlayer hybrid composites: (**a**) [G/G/C/G]; (**b**) [G/G/G/C].

*3.4. Failure Behaviors of Interlayer and Intralayer Hybrid Composites with the Hybrid Ratio C:G = 1:4*

Figure 12 shows that in the tensile force–displacement curves of hybrid composites compared with the hybrid ratios C:G = 1:1, 1:2, 1:3, and above, the second-order failure force of hybrid composites with C:G = 1:4 was the closest to the first-order failure force. Additionally, the failure force of the symmetric structure with glass fiber sandwiching carbon fiber [G/G/C/G/G] was high and the second failure did not appear with the assistance of the synergistic effect. For the asymmetric structure, the failure force was low and the second failure process lasted longer. Additionally, the tensile of the intralayer structure with high dispersion degree was small.

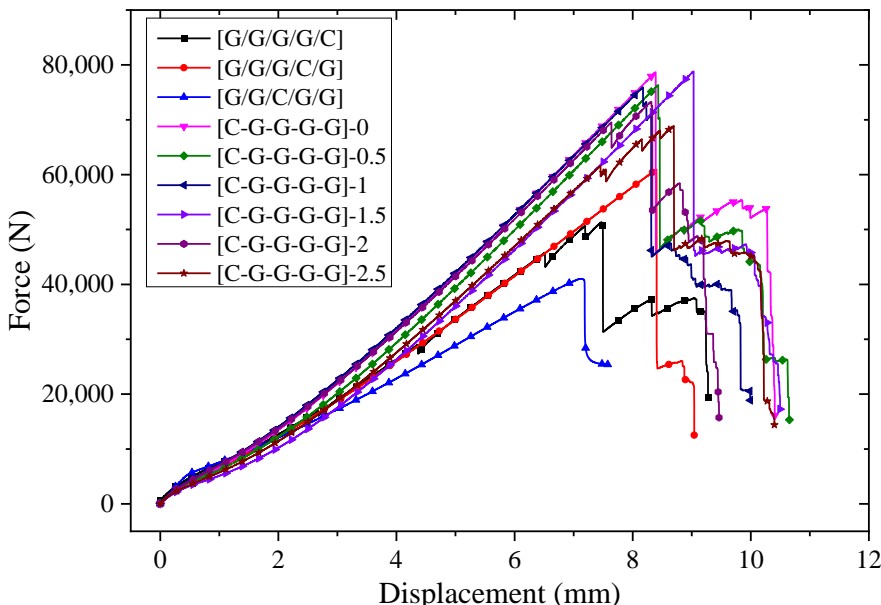

**Figure 12.** Tensile force–displacement curves of interlayer and intralayer composites with the hybrid ratio C:G = 1:4.

Figure 13a displays the failure process of [G/G/G/G/C]. After the first failure, the carbon layer was split; however, the glass layer still maintained a high level of integrity, with low damage caused by the carbon fiber. As the tension force continued to increase, the glass layer finally broke. Figure 13b shows the failure process of [G/G/G/C/G]. We observed significant damage to the glass fiber near the carbon fiber on the sample surface, while the opposite side experienced minimal damage. Figure 13c presents the failure process of [G/G/C/G/G]. Since two layers of glass fiber were distributed on either side of the carbon layers, after the first failure of the carbon fiber, no apparent surface damage was observed. However, an internal collapse failure of the glass fiber exhibited was evidenced by an obvious white color. Figure 13d shows the failure process of intralayer hybrid composites [C/G/G/G/G]-0.5, where carbon fiber failure only occurred in one place; furthermore, due to its low content, the failure had a limited effect on the glass fiber.

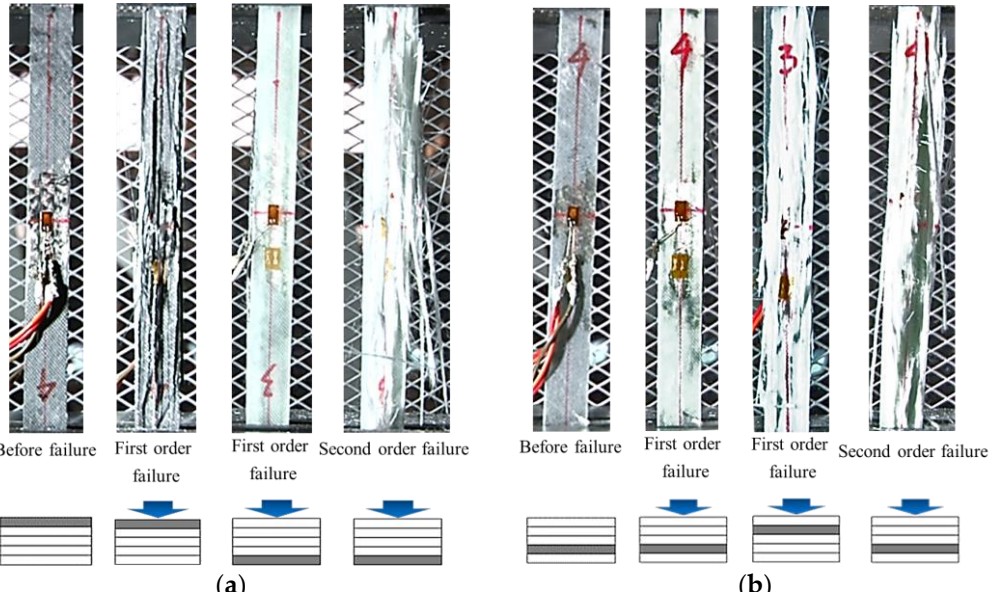

**Figure 13.** *Cont.*

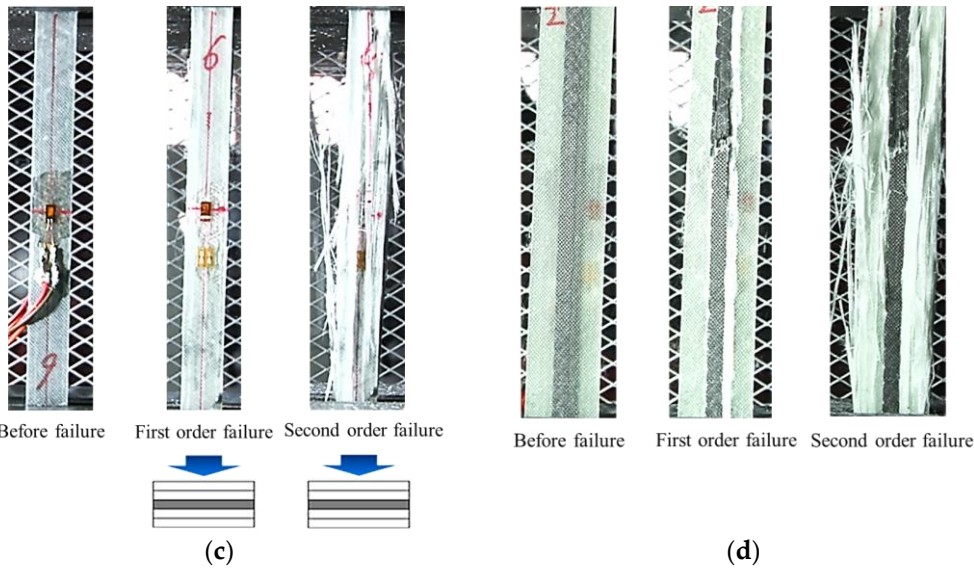

**Figure 13.** Failure process of hybrid composites: (**a**) [G/G/G/G/C]; (**b**) [G/G/G/C/G]; (**c**) [G/G/C/G/G]; (**d**) Intralayer composites [C/G/G/G/G]-0.5.

### 3.5. Failure Theory of Hybrid Composites

#### 3.5.1. Synergistic Effect

Hybrid composites are composed of two fiber reinforcements, assuming a lack of friction between them, while subjected to tension, the material with a low fracture strain will break and fail first, followed by the failure of the higher-strain material. Typically, the damage of the two materials occurs independently with no mutual effect. However, due to the bonding effect of the resin at the interface of the two materials, this ideal failure is unlikely. In practice, the force transfers from one material to another, and mutual assistance distributes the load, preventing mutual failure. This phenomenon is known as the synergistic effect. The degree of the synergistic effect depends on the hybrid structure and ratio, with a greater synergistic effect resulting in better mechanical properties. The tensile properties of various hybrid composites can be compared to evaluate the degree of synergistic effect.

For C/G hybrid composites, carbon fibers tend to become damaged earlier than glass fibers, due to their lower breaking strain. However, glass fiber would hinder the damage of carbon fiber. Figure 14a depicts the failure process of an interlayer hybrid structure with a hybrid ratio C:G = 1:4, the white zone within the red frame indicates delamination failure within the specimen due to the force differences between carbon fiber and glass fiber. In the experiment, before the first-order failure of the hybrid composites, a tiny cracking sound was heard within the fiber, this was found to be small. amount of delamination at the C/G interface. As the tensile load continued to increase, the delamination area progressively expanded, and as the specimen approached the first-order failure, the delamination area occupied most of the sample. A similar phenomenon was observed at the C/G interface in the intralayer hybrid composites shown in Figure 14b, however, this failure was not as evident as in the interlayer hybrid structures.

Delamination at the C/G interface in Figure 14a,b can mainly be attributed to the synergistic effect of hybrid composites. The failure mechanism of the structure with glass fiber sandwiching carbon fiber is exhibited in Figure 15. During tension, as the tensile strain approaches the breaking strain of carbon fiber, it breaks in many locations. Whereas the interlaminar shear force at both sides of the carbon fiber is exerted from the external glass fiber, the glass fiber assumes a part of the load and delays the failure process, thus preventing the crack propagation of carbon fiber composites. In this structure, glass fiber exhibits a certain amount of synergistic effect on the composites, enhancing the bearing capacity of the hybrid composites.

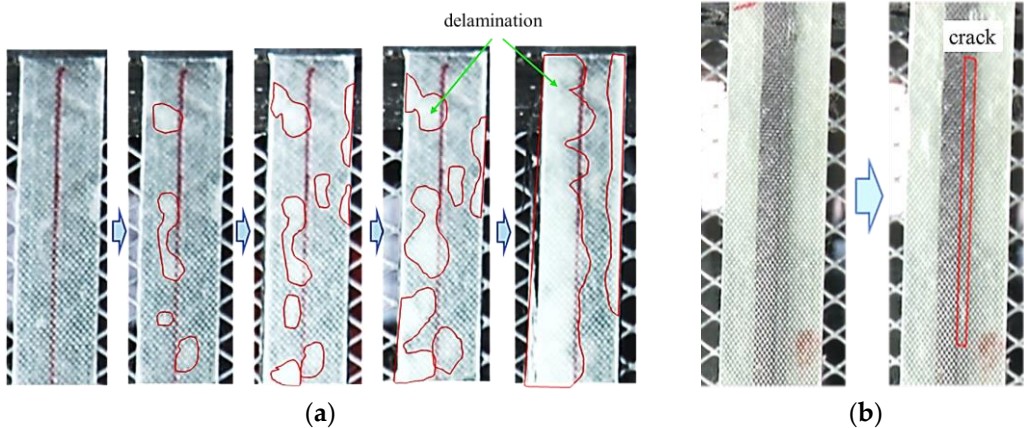

**Figure 14.** Tensile failure process of hybrid composites: (**a**) the interlayer hybrid structure with C:G = 1:4; (**b**) failure process of intralayer hybrid composites.

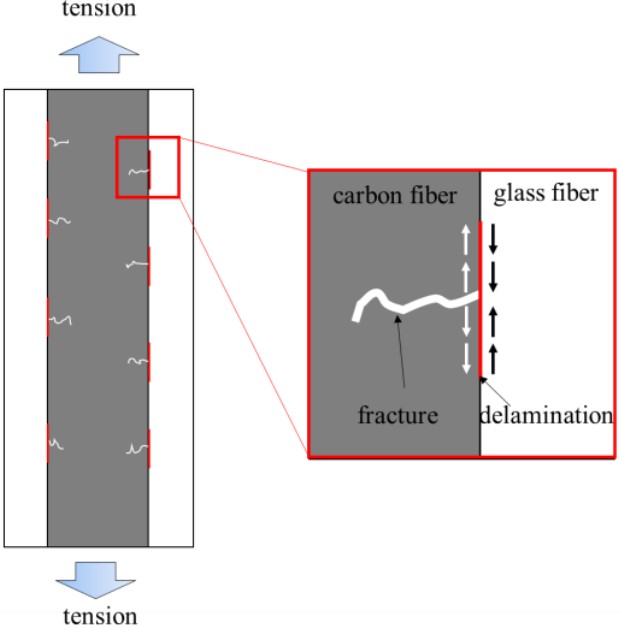

**Figure 15.** Schematic diagram of the synergistic effect for the structure with glass fiber sandwiching carbon fiber.

The fracture zone of C/G interface exhibits micro delamination in the red line in Figure 15, which is attributed to the fracture of carbon fiber and the shear force. This phenomenon is also observed in the failure process shown in Figure 14. To exert the synergistic effect of glass fiber on carbon fiber, the carbon fiber needs to be located optimally within the composites laminate. In this configuration, the glass fiber on both sides of the carbon fiber limited the propagation of cracks of the carbon fiber. However, if the carbon fiber is distributed at the surface layer, the inward glass fiber only provides a load for the external carbon fiber at the C/G bonding zone. As a result, there is no synergistic effect on the other side of the carbon fiber, causing early failure of the carbon fiber and resulting in the structure with carbon fiber sandwiching glass fiber or the asymmetric structure with low tensile strength.

### 3.5.2. Damage Acceleration Effect

The synergistic effect between two materials is characterized by their mutual reinforcements, typically seen in hybrid composites. Glass fiber provides a synergistic effect for carbon fiber and hinders its destruction. However, a damage acceleration effect in the failure process of hybrid composites may exist, which is evident in Figure 16. Figure 16a shows the failure theory of the structure with glass fiber sandwiching carbon fiber. Upon applying the tension, the carbon fiber is destroyed first, causing the first-order failure, and almost all the damage energy transfers to the outer glass fiber layer, causing considerable damage and accelerating its failure. Consequently, second-order failure occurs rapidly. Figure 16b illustrates an asymmetric interlayer hybrid structure where only one side of the carbon fiber layer is clamped by the glass fiber. In this case, a lower amount of the carbon fiber's damage energy transfers to the glass fiber on the other side, causing less damage to the glass fiber; therefore, the stacking structure has a significant impact on the failure process of C/G hybrid composites. For the case of glass fiber sandwiching carbon fiber, the damage acceleration is predominantly caused by the internal carbon fiber, resulting in a more severe damage process for the composites. Conversely, as carbon fiber distributes on one side or sandwiches the glass fiber, the failure process becomes relatively longer.

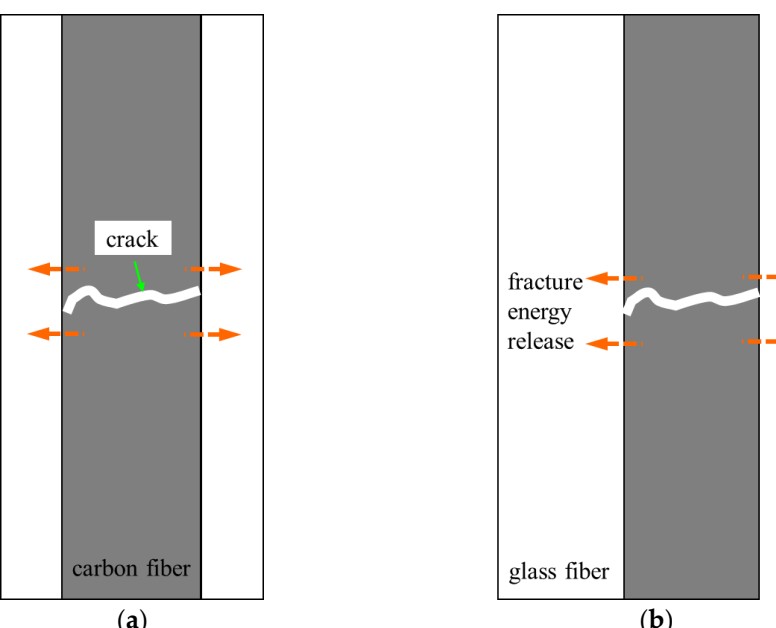

**Figure 16.** Fracture energy release outlet of interlayer hybrid composites: (**a**) glass fiber sandwiching carbon fiber; (**b**) asymmetric carbon/glass hybrid structure.

### 3.5.3. Stress–Strain Decay of Hybrid Composites

According to failure synergistic theory and failure acceleration theory, the actual stress–strain curves of hybrid composites appear to differ from the theoretical ones. Figure 17 presents the theoretical curves of hybrid composites, which suggest that in an ideal failure process of C/G hybrid composites, as the hybrid composites reach the breaking strain of carbon fiber, the carbon fiber fails, the tensile force drops sharply, and the first-order failure occurs, with glass fiber merely assuming the residual load until damage; thus, the theoretical stress–strain of C/G hybrid composites presents a zigzag shape. However, during the experiment, as the elongation approached the failure strain of the carbon fiber composites, delayed the fracture of the carbon fiber and made the tensile strength exceed the theoretical value. However, instead of breaking simultaneously, the carbon fiber underwent only gradual failure. Once it was completely broken, the sample experienced a violent first-order failure. The high fracture energy released by the carbon fiber is then transferred to the glass fiber, causing a strength loss in the glass fiber to some extent. As a result,

the first failure strength of the hybrid composites surpassed the theoretical value, but the second-order failure force was lower, indicating an accelerating failure.

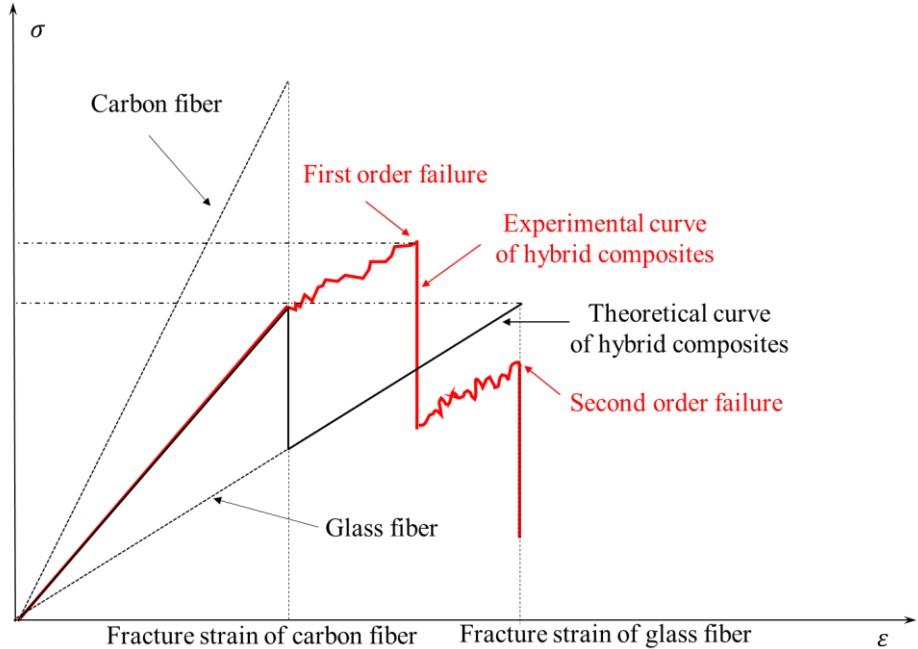

**Figure 17.** Tensile stress–strain curves of C/G hybrid composites.

## 4. Tensile Properties of Interlayer and Intralayer Hybrid Composites

### 4.1. Tensile Properties of Interlayer Hybrid Composites

In this study, the tensile properties of C/G hybrid composites were attained through testing. Figure 18 shows the tensile modulus, strength, and fracture strain of interlayer hybrid composites with various hybrid ratios and structures. This section presents the effect of the hybrid ratio and structure on the tensile properties of interlayer hybrid composites.

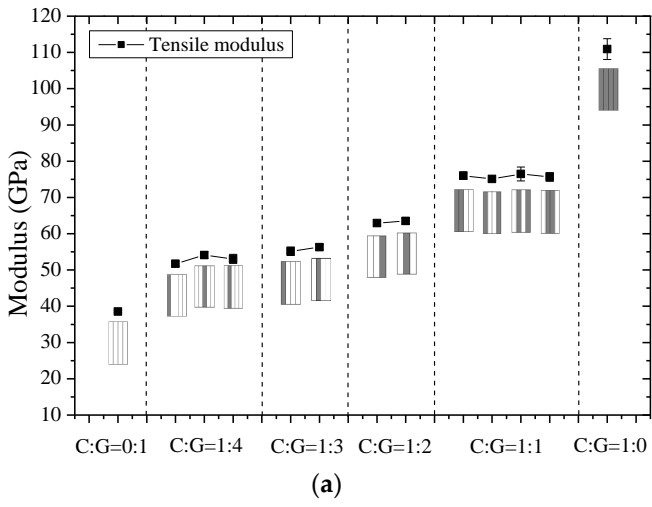

(a)

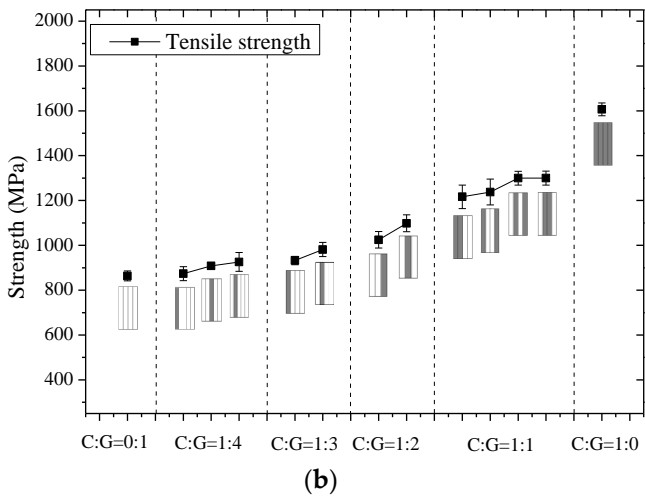

(b)

**Figure 18.** *Cont.*

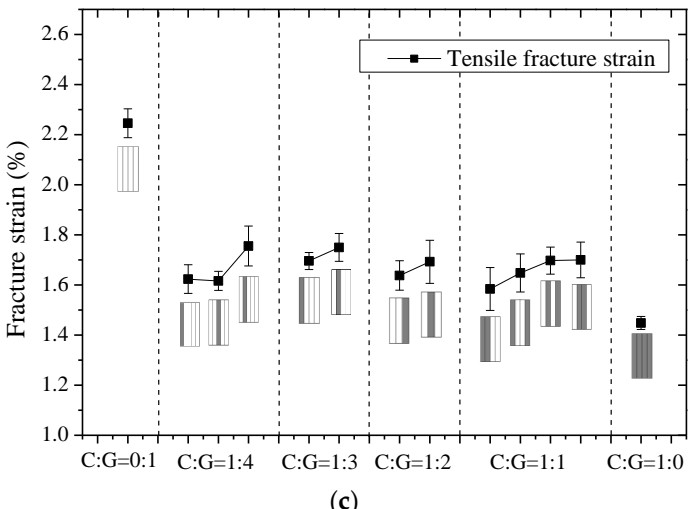

**Figure 18.** Tensile properties of interlayer hybrid composites with different hybrid ratios and stacking structures: (**a**) tensile modulus; (**b**) tensile strength; (**c**) fracture strain.

Figure 18a displays the tensile modulus of interlayer hybrid composites. We found that, with the increase in carbon fiber content, the tensile modulus increased gradually, and the value fell between that of the carbon fiber and glass fiber. When the hybrid ratio remained constant, the tensile modulus of the various stacking structures showed minor fluctuations that were below 5%. Consequently, the decisive factor for the tensile modulus of the interlayer hybrid composites was revealed to be the hybrid ratio; however, it was also related to the stacking structure.

Figure 18b shows the tensile strength of the interlayer hybrid composites with different hybrid ratios and stacking structures. An increase in carbon fiber content can enhance the tensile strength of interlayer hybrid composites, and the values fall within the range of those of pure carbon fiber and glass fiber. As the hybrid ratio was the same, the structures with glass fiber sandwiching carbon fiber, such as [G/G/C/G/G], [G/G/C/G], [G/C/G], and [G/C/C/G], displayed 6%–8% higher tensile strength than the structures with carbon fiber and glass fiber distributed asymmetrically, inclusive of [C/G/G/G/G], [C/G/G/G], [C/G/G], and [C/C/G/G], as observed in Li et al.'s report [25]. The primary reason for the result was the structure with glass fiber sandwiching carbon fiber, with the glass fiber on both sides providing a certain synergistic effect for the inward carbon fiber, hindering the crack expansion of the carbon fiber layer and resulting in the composite's relatively high strength. On the contrary, with carbon fiber distributing on one side or carbon fiber sandwiching glass fiber, the carbon fiber failed first, with the glass fiber on one side or in the core providing a weak synergistic effect towards the surface of the carbon layer, resulting in lower tensile strength. In conclusion, the hybrid ratio and stacking structure played a decisive factor in determining the tensile strength of interlayer hybrid composites.

Figure 18c shows the fracture strain of interlayer hybrid composites. With the increase in carbon fiber content, the fracture strain of the interlayer hybrid composites did not appear to vary, apparently remaining at a constant level. This was because the tensile strength reached the maximum value after the first-order fracture failure of the carbon fiber, and the broken strain of the hybrid composites was the strain of the carbon fiber; however, due to the existence of a synergistic effect, the first-order fracture strain of the various stacking structures was slightly different. Moreover, under the same hybrid ratio for structures with glass fiber sandwiching carbon fiber, inclusive of [G/C/G/G/G], [G/C/G/G], [G/C/G], and [G/C/C/G], the fracture strain of the composites was high, consistent with the effect of tensile strength. Conversely, in structures with asymmetric carbon fiber and glass fiber

distribution, such as [C/G/G/G/G], [C/G/G/G], [C/G/G], and [C/C/G/G], the failure strain tended to be low.

In summary, the hybrid ratio C/G played a key factor in the tensile strength of interlayer hybrid composites; however, it had little effect on the fracture strain and modulus. With the same hybrid ratio, the stacking structure had a certain effect on the tensile modulus, strength, and strain, which was primarily applicable to the hybrid structure with glass fiber sandwiching carbon fiber, and the synergistic effect between carbon fiber and glass fiber was evident, whereas it could improve the tensile strength. On the contrary, for other structures, the synergistic effect was weak, which resulted in the tensile strength and strain of the interlayer hybrid composites being low.

### 4.2. Tensile Properties of Intralayer Hybrid Composites

This section analyzes the tensile properties of intralayer hybrid composites with various hybrid ratios and stacking structures. Figure 19 exhibits the tensile modulus, strength, and fracture strain of intralayer hybrid composites.

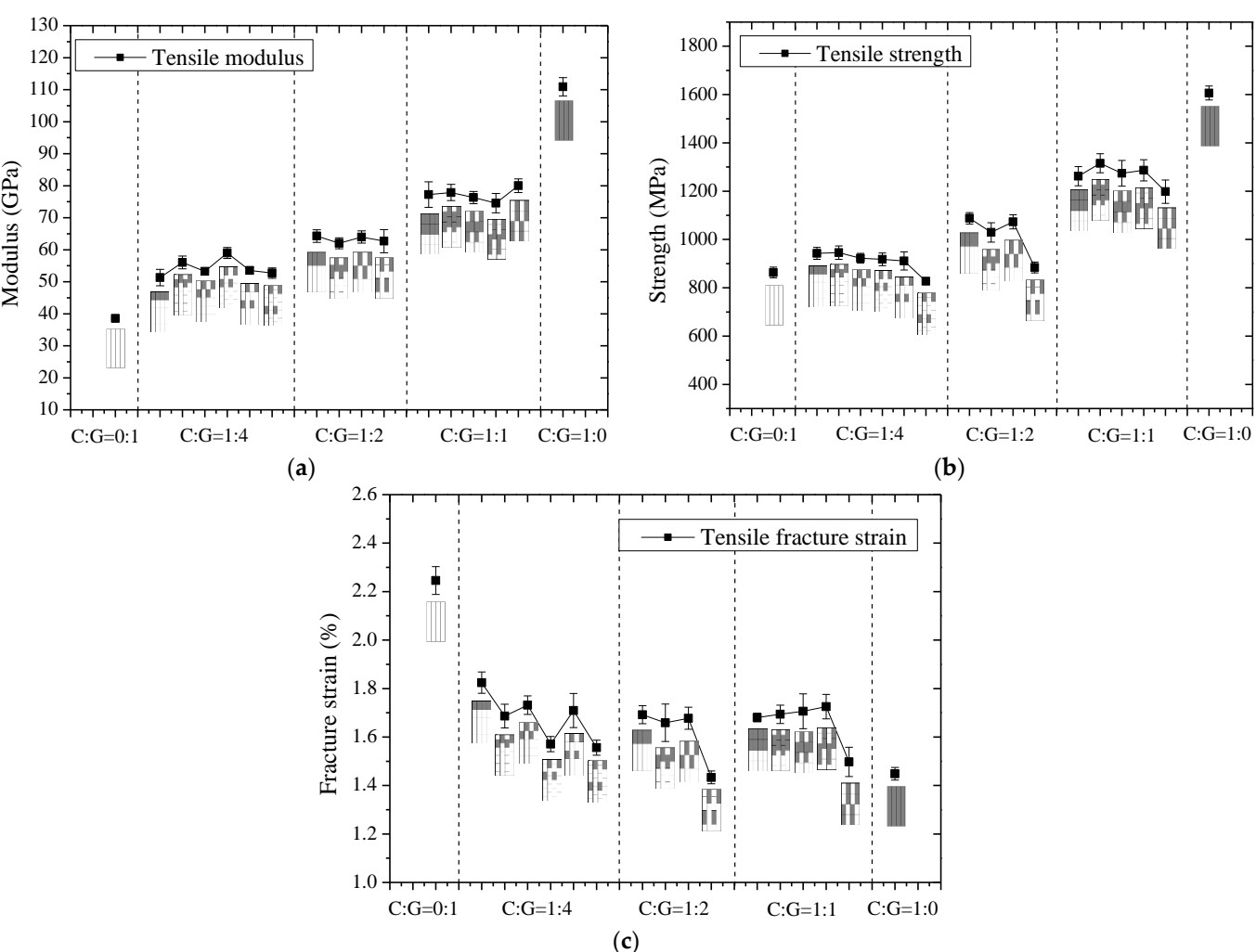

**Figure 19.** Tensile properties of intralayer hybrid composites with various hybrid ratios and stacking structures: (**a**) tensile modulus; (**b**) tensile strength; (**c**) fracture strain.

Figure 19a shows that the tensile modulus of intralayer hybrid composites increased progressively with the increase in carbon fiber content, and its fall between carbon fiber and glass fiber. It is worth noting that changes in intralayer hybrid structures at the same hybrid ratio exerted no evident impact on the tensile modulus.

Figure 19b depicts the tensile strength of intralayer hybrid composites. With the increase in carbon fiber content, the tensile strength displayed an increasing trend, and it maintained between carbon fiber and glass fiber. Under the same hybrid ratio, as the dispersion degree of the intralayer laminate was high, the tensile strength was low, and the strength of [C-G-G-G-G]-2.5 even decreased by 4% compared to that of glass fiber. Meanwhile, the tensile strength of [C-C-G-G]-2 with C/G = 1:2 also remained at the same level of glass fiber, and a similar trend was applicable for the fracture strain of intralayer hybrid composites, as shown in Figure 18c. While the dispersion degree was high, the fracture strain tended to decline, which was related to the synergistic effect.

Table 8 shows the cutting diagrams of intralayer hybrid composites with various hybrid ratios and structures. With the same hybrid ratio, as the dispersion degree increased, such as in [C-G]-1, [C-C-G-G]-2, and [C-G-G-G-G]-2.5, the sample side distributed carbon fiber, resulting in a weak synergistic effect between carbon fiber and glass fiber, resulting in lower tensile strength and strain. Meanwhile, for other structures, the glass fibers were primarily distributed throughout the sample, leading to a more effective synergistic effect, resulting in higher tensile strength and strain.

**Table 8.** Schematic diagram of cutting scheme for intralayer hybrid composites with various hybrid ratios and stacking structures.

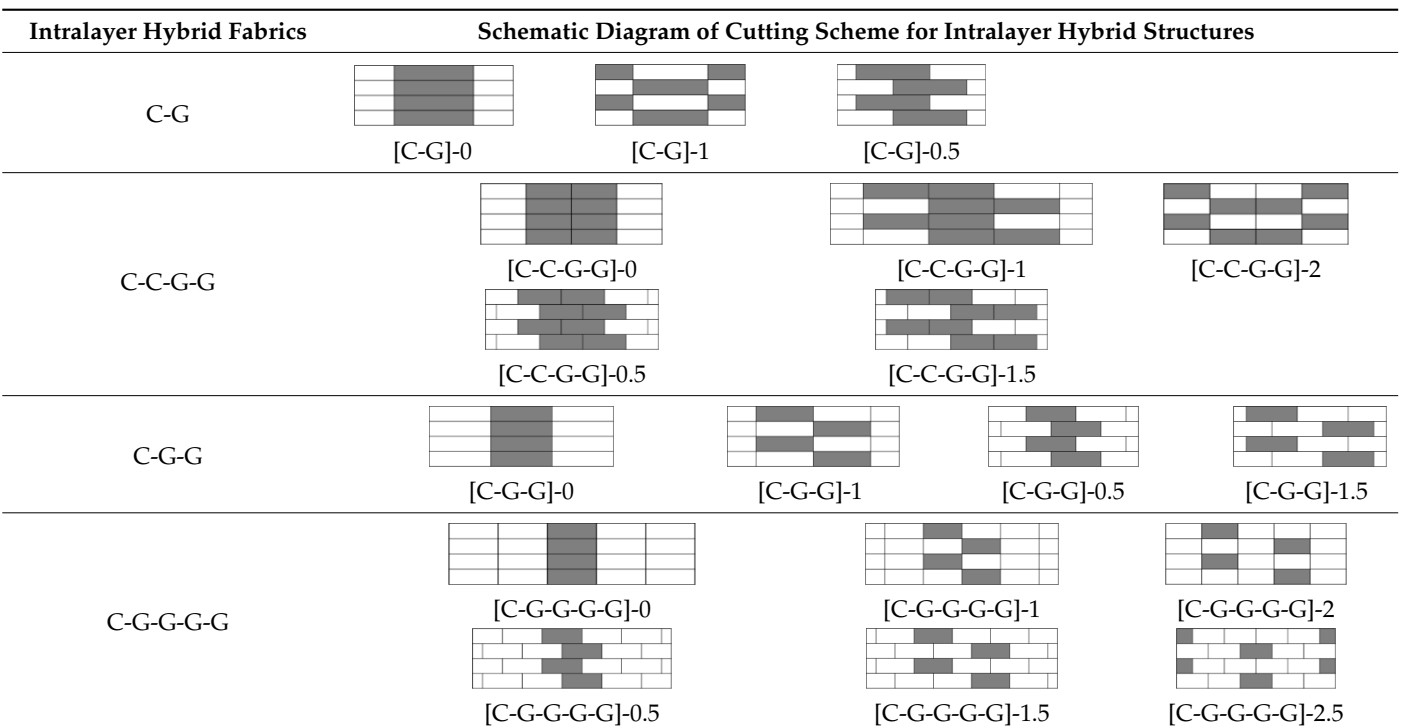

For C:G = 1:1, Figure 20 shows a comparison of the tensile properties of the intralayer hybrid structures [C-G] and [C-C-G-G]. We found that the tensile modulus and strength were maintained at the same level. As the dispersion degree increased, such as in [C-G]-1 and [C-C-G-G]-2, the tensile strength exhibited a decrease.

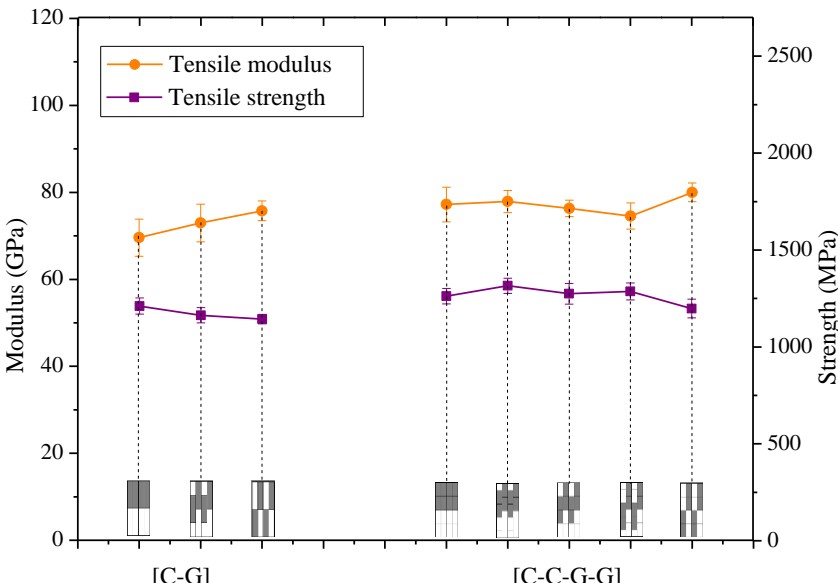

**Figure 20.** Comparison of tensile modulus and strength of intralayer hybrid structure with C:G = 1:1.

Based on the analysis presented, it can be concluded that the hybrid ratio was regarded as the decisive factor for the tensile modulus and strength of intralayer hybrid composites. An increase in carbon fiber content can increase the tensile modulus and strength of intralayer hybrid composites; however, the fracture strain was determined by the layup sequences. Additionally, under the same hybrid ratio, the tensile strength and fracture strain of intralayer hybrid composites were still impacted by the stacking structure. In general, the high dispersion degree of carbon fiber and glass fiber would weaken the strength and fracture strain due to the weak synergistic effect of the C/G interface.

*4.3. Comparison of Theoretical and Experimental Tensile Strength forInterlayer and Intralayer Hybrid Composites*

In general, two reinforcements in hybrid composites have different mechanical properties, which may enhance or weaken the properties of composites. To evaluate the hybrid effect and whether the tensile properties of hybrid composites change, the rule of the mix (ROM) was introduced. Hybrid tensile modulus is to calculate the tensile modulus of hybrid composites based on the hybrid ratio of two fibers and their respective tensile modulus [26], the formula is as follows:

$$E_{TROM} = V_C E_{TC} + V_G E_{TG} \tag{1}$$

Among:

$E_{TROM}$ : Hybrid tensile modulus derived from ROM (GPa);
$E_{TC}$: Tensile modulus of carbon fiber (GPa);
$E_{TG}$: Tensile modulus of glass fiber (GPa);
$V_C$: The fiber volume fraction of carbon fiber composites;
$V_G$: The fiber volume fraction of glass fiber composites.

Figure 21 shows a comparison between the experimental and calculated tensile modulus of interlayer and intralayer hybrid composites. It was found that the tensile modulus of interlayer and intralayer hybrid composites increased with the increase of carbon fiber content and exhibited a linear relationship, furthermore, the experimental values were slightly higher than theoretical ones, presenting a weak positive hybrid effect.

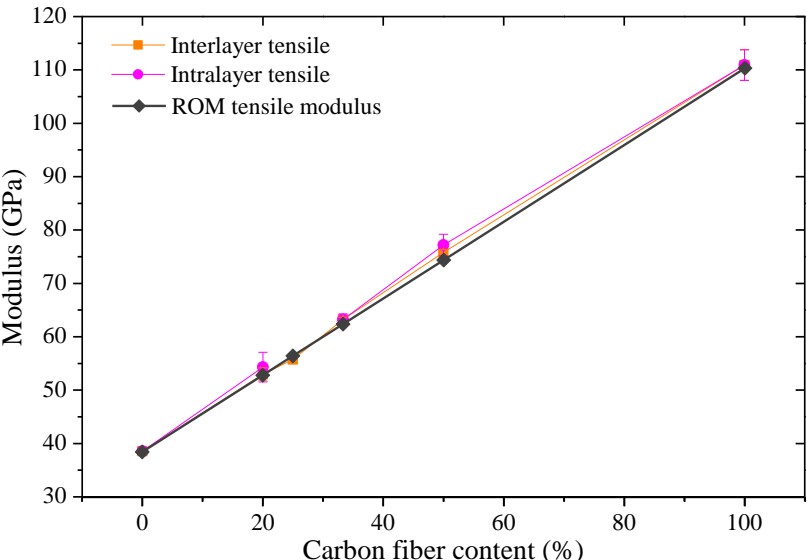

**Figure 21.** Comparison of experimental and theoretical tensile modulus.

Due to the difference in tensile fracture strain for carbon fiber and glass fiber, carbon fiber breaks earlier than glass fiber, therefore, the tensile strength of hybrid composites cannot be calculated by ROM, the strength of carbon fiber after fracture should be determined by the content of glass fiber. If the glass fiber content is high, glass fiber will continue to bear the load, if the glass fiber content is small, the remaining glass fiber can't bear the load leading the sample to damage quickly [33]. Meanwhile, the calculation of tensile strength for hybrid composites adopted Formulas (2) and (3) [34]:

$$\text{Before carbon fiber fracture } \sigma_{HY} = (V_C E_C + V_G E_G)\varepsilon \tag{2}$$

$$\text{After carbon fiber fracture } \sigma_{HY} = V_G E_G \varepsilon \tag{3}$$

Among

$\sigma_{HY}$: The stress of hybrid composites (MPa)

$V_{fC}$ : The fiber volume fraction of carbon composites;

$V_{fG}$ : The fiber volume fraction of glass composites;

$E_C$: Tensile modulus of carbon fiber (GPa);

$E_G$: Tensile modulus of glass fiber (GPa);

$\varepsilon$: The strain of hybrid composites.

Figure 22 shows a comparison of experimental and theoretical tensile strength of interlayer and intralayer hybrid composites. It was found that the theoretical value of hybrid composites exhibited a trend of first decrease and then an increase. The tensile strength of hybrid composites increased with the increase of carbon fiber content, and the experimental strength was greater than the theoretical value, which indicated tensile strength of hybrid composites presented a positive hybrid effect. Compared tensile strength of interlayer and intralayer hybrid composites with various hybrid ratios and structures, it was found that the of the strong dispersion of intralayer hybrid composites caused by the stacking structure was higher than that of interlayer hybrid composites.

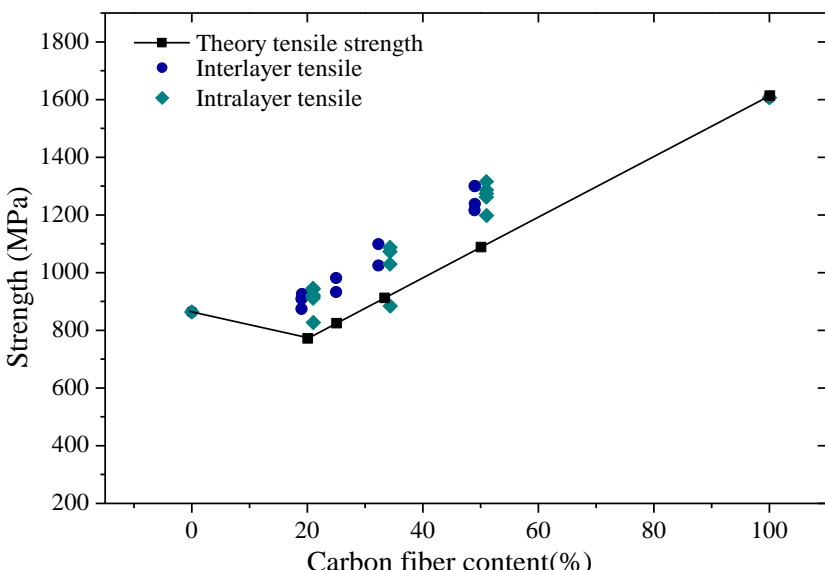

**Figure 22.** Theoretical and experimental strength of interlayer and intralayer hybrid composites.

## 5. Conclusions

In this study, carbon/glass non-crimp-fabric-reinforced interlayer and intralayer hybrid composites were designed systematically, the tensile properties and failure behaviors of hybrid composites were investigated, and the following conclusions can be drawn:

The damage process analysis of hybrid composites revealed that with the assistance of the synergistic effect, the first-order tensile strength could reach the maximum value for the structure with glass fiber sandwiching carbon fiber, the failure of the specimen was the most apparent, and the second-order failure process remained short due to the failure acceleration effect. However, in cases where the laminate structure was asymmetric, the second-order failure process remained for a relatively long time. Through the analysis of the failure process, the proposed synergistic effect and damage acceleration effect could be utilized to explain tensile properties and failure for hybrid composites.

Comparisons of the tensile properties of interlayer and intralayer hybrid composites revealed that the tensile strength of the interlayer and intralayer hybrid composites increased with the carbon fiber content, and the excellent tensile strength of the interlayer hybrid structure with glass fiber sandwiching carbon fiber can be obtained at the same hybrid ratio. In intralayer composites with high hybrid degrees, the tensile strength and strain were relatively low. The fracture strain of both interlayer and intralayer hybrid composites was generally low, equivalent to that of carbon fiber. Compared to the experimental and theoretical modulus and strength of interlayer and intralayer hybrid composites, it was found they presented a positive hybrid effect, and the experimental values were all superior to the theoretical values. This study paves the way for optimizing the properties of C/G NCF hybrid composites while keeping costs low, thereby opening new possibilities for their applications.

**Funding:** This research was funded by the Scientific Research Foundation of Zhejiang Sci-Tech University (Grant No. 11152932612007); Foundation for Excellent PH.D. Program Zhejiang Sci-Tech University (Grant No. 11150131722008); and Foundation for Youth Innovation of Zhejiang Sci-Tech University (Grant No. 11152931632104); Scientific Research Fundation of Zhejiang Provincial Education Department: (Grant No. 11152832622207).

**Institutional Review Board Statement:** Not applicable.

**Informed Consent Statement:** Not applicable.

**Data Availability Statement:** All relevant data generated by the authors during the study are included within the paper.

**Conflicts of Interest:** The author declares no conflict of interest.

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
