# Peer review of "Tensile Failure Behaviors and Theories of Carbon/Glass Hybrid Interlayer and Intralayer Composites"

_coatings, doi:10.3390/coatings13040774_

Round 1

Reviewer 1 Report

This study deals with "Tensile failure behaviors and theories of carbon/glass hybrid interlayer and intralayer composites". The manuscript has enough innovation and the result provides useful data. The authors must check these comments:

The abstract must be written more scientifically. General information is provided. It needs to be revised.

It seems that some references do not match the existing text and are confusing. A detailed review is needed. for example:

Page 2 line 46: "Martin [16]" is written. In Ref. 16, it is not the author's name Martin. Is there a mistake? check it.

Page 2 line 48: It should be written: "Zeng et al."

page 2 line 65: Kedar [23], Does not match the reference.

page 2 line 69: Ebrahimnezhad et al. [25] instead of Hossein [25].

page 2 line 75: Karahan et al. [27] instead of Karahan [27]

and so on...

The authors must summarize the last paragraph of the introduction. Just explain exactly what is discussed in the following sections in a scientific way.

In sections 3 and 4 of the manuscript (3- Tensile Process and Failure Behaviors of Interlayer and Intralayer Hybrid Composites & 4. Tensile Properties of Interlayer and Intralayer Hybrid Composites), there is almost no comparison with other literature. Therefore, it is not possible to know their authenticity. Authors are strongly advised to properly compare their results with other literature.

Author Response

Dear reviewer 1:

Thank you for your valuable time and effort in reviewing my article, Please see the attachment.

Reviewer 2 Report

This is an interesting study about Tensile failure behaviors of  intralayer composites. Yet, there are some typos and English writing problems in the manuscript, which should be considered by the authors for publication. The authors should also consider the following comments for their manuscript.

1--The introduction is not sufficient. The authors must expand the introduction section and use the most recent studies in this area and application of composite structures (carbon/graphene) in other areas of engineering. Following studies is recommended to use in introduction.

1-https://doi.org/10.1177/1077546320923930

2-https://doi.org/10.1080/15397734.2019.1705166

4-https://doi.org/10.1016/j.compstruct.2021.114438

2- In Materials and Methods section more explanation is required.

3-  In Figures 4-7-9 and 11 authors are encouraged to use pictures with higher quality that can demonstrate the failure process in a better way.

4-more justification is required in the conclusions section.

5- references is not sufficient. The authors are required to use the most recent studies including the application of novel composite structures in other areas of engineering. 

Author Response

Dear reviewer:

Thank you for taking the time and effort to review my article. Please see the attached.

Reviewer 3 Report

There are some weaknesses through the manuscript which need improvement. Therefore, the current study cannot be accepted for publication in this form, but it has a chance of acceptance after a major revision. My comments and suggestions are as follows:

1- Abstract gives information on the main feature of the performed study, but some details about the conducted tests in this study must be added.

2- Authors must clarify necessity of the performed research. Objectives of the study must be clearly mentioned in introduction.

3- The literature study must be enriched. In this respect, authors must read and refer to the following papers: (a) https://doi.org/10.1016/j.compstruct.2018.05.043 (b) https://doi.org/10.1016/j.tws.2022.109515 and other relevant research works.

4- Novelty needs to be explicit in highlights and abstract.

5- Broaden and update the literature review to better connect to the current effort in the field in the context of fracture in composites. Authors can search in ScienceDirect.

6- Why this particular material is used in this study.

7- It is necessary to summarize all obtained values of experiments in a table.

8- All figures must be illustrated in a high quality. Some legends in figures (curves) are illegible. In addition, all images must be more informative. Also, the scale bar is required in some figures. 

9- Details of calculation of modulus must be added to the paper.

10- In its language layer, the manuscript should be considered for English language editing. There are sentences which have to be rewritten.

11- The conclusion must be more than just a summary of the manuscript. List of references must be updated based on the proposed papers. Please provide all changes by red color in the revised version.

Author Response

(The authors gave the same response as above.)

Round 2

Reviewer 1 Report

It seems that the manuscript is ready for publication.

Reviewer 2 Report

Congratulations, your current paper is now accepted.

Reviewer 3 Report

The paper has been improved and corresponding modifications have been conducted. In my opinion, the current version can be considered for publication.